# Sufficiency of the BOT-2 short form to screen motor competency in preschool children with strabismus

Kuo-Kuang Yeh[1,2]☯, Wen-Yu Liu[1,2‡]*, Meng-Ling Yang[3‡], Chun-Hsiu Liu[3‡], Hen-Yu Lien[2‡], Chia-Ying Chung[1‡]

**1** Department of Physical Medicine and Rehabilitation, Chang Gung Medical Foundation, Linkou Chang Gung Memorial Hospital, Taoyuan City, Taiwan, **2** School of Physical Therapy and Graduation Institute of Rehabilitation Science, College of Medicine, Chang Gung University, Taoyuan City, Taiwan, **3** Department of Children Ophthalmology, Chang Gung Medical Foundation, Linkou Medical Center, Taoyuan City, Taiwan

☯ These authors contributed equally to this work.
‡ WYL, MLY, CHL, HYL and CYC also contributed equally to this work.
* wylpt@mail.cgu.edu.tw

## Abstract

### Background

Strabismus is one of the most common visual disorders in children, with a reported prevalence of 2.48% in preschoolers. Additionally, up to 89.9% of preschool children with strabismus do not have normal stereopsis. Whether this lack of normal stereopsis affects the motor competency of preschool children with strabismus is unknown. The Bruininks-Oseretsky Test of Motor Proficiency Second Edition short form (BOT-2 SF) can be a useful tool for screening; however, its sufficiency as a diagnostic tool for children with various disorders is controversial.

### Objective

The aims of this study were thus to examine motor competency in preschool children with strabismus by using the BOT-2 and to evaluate the usefulness of the BOT-2 SF to identify those at risk for motor competency issues.

### Methods

Forty preschool children (aged 5–7 years) with strabismus were recruited, all of whom had abnormal stereopsis. The BOT-2 complete form (CF) was administered to all children. The BOT-2 CF was administered to all children. The scores of the BOT-2 SF were extracted from the relevant items of the BOT-2 CF for further analysis.

### Results

The prevalence of children with strabismus who had below average performance in the composites of "Fine Manual Control", "Manual Coordination","Body Coordination", and "Strength and Agility" were 15%, 70%, 32.5%, and 5%, respectively, on the BOT-2 CF.

**Data Availability Statement:** All relevant data are within the manuscript and its Supporting Information files.

**Funding:** The funders had no role in study design, data collection and analysis, or decision to publish, but provided partial financial support for the English editing after the completion of first draft of this manuscript. Chang Gung Memorial Hospital, Linkou provided funding to WYL (award number: BMRP652).

**Competing interests:** The authors have declared that no competing interests exist.

Compared with these results, the sensitivity of the BOT-2 SF was 33.33% (95% CI = 7.49%–70.07%) and the specificity was 100% (95% CI = 88.78%–100%).

## Conclusion

Preschool children with strabismus had a high prevalence of impaired motor competency, especially in fine motor competency. The BOT-2 SF was not as sensitive in identifying motor difficulties in preschool children with strabismus. Therefore, the BOT-2 CF is recommended for evaluating motor proficiency in preschool children with strabismus.

## Introduction

Strabismus is defined as a deviation of the eye from perfect ocular alignment and can be further divided into esotropia and exotropia, or less commonly, hypertropia and hypotropia [1]. It is one of the most common visual development disorders with a reported global prevalence of 1.93% (95% CI = 1.64%–2.21%) [2]. For preschool-aged children, the prevalence of strabismus ranges from 1.5% to 3.8% [3–5]. Preschool children with strabismus have abnormal stereopsis [6] and are at greater risk of amblyopia [7], which is associated with abnormal binocular vision [8].

Additionally, each of those conditions potentially affects the development of motor competency. The motor competency of children with strabismus has been examined in various age groups including infants [9–11], school-aged children (7–15 years old) [6, 12–14], and adolescents (>15 years old) [15, 16] (Table 1). Despite development of motor competency at preschool age being critical for the overall development of individuals, information regarding the motor competency of preschool children with strabismus is limited. Motor competency is multidimensional, ranging from simple to combined to complex, which are all inter-related, and may be further divided into gross motor competency and fine motor competency in standardized assessments [17, 18]. However, previous studies on children with strabismus tend to focus more on a single aspect, either gross motor competency [10, 11, 13, 16], or fine motor competency [6, 12, 15], rather than complete motor competency [14, 19] (Table 1). Therefore, it is necessary to investigate the complete motor competency of preschool children with strabismus.

Before conducting a comprehensive assessment of motor competency, children must be efficiently screened using developmental tools to identify those with strabismus who are suspected to have below age-appropriate motor competency. The Bruininks–Oseretsky Test of Motor Proficiency-Second Edition (BOT-2) [20] is a well-known measure of motor proficiency designed to provide clinicians with useful motor competency information in children. The BOT-2 can be utilized in a complete form (CF) or short form (SF). Discriminative validity, inter-rater, and test-retest reliabilities are good between the two forms, as reported in the clinical studies: for developmental coordination disorder; high-functioning autism/Asperger's disorder; and mild to moderate intelligent disability [21]. However, Mancini and colleagues (2020) reported that the BOT-2 SF overestimates a child's motor proficiency relative to the BOT-2 CF in school-age children with attention deficit hyperactivity disorder (ADHD) [22]. Chung and colleagues (2012) reported an elevated ADHD symptom rate of 15.7% in children with strabismus and concluded that childhood strabismus might be mistaken as ADHD-related symptoms [23]. Children with strabismus have shown an increased risk (OR = 2.13, 95% CI = 1.52–2.99) for ADHD [24]. Considering the relevance of childhood strabismus to ADHD-related symptoms, whether the BOT-2 SF is effective in testing the motor competency

**Table 1. Previous motor competency studies on subjects with strabismus.**

| Studies | | Present study | Hemptinne et al. | Vagge et al. | Zipori et al. | Di Sipio et al. | O'Connor et al. | Webber et al. | Drover et al. | Tukkers-van et al. | Caputo et al. | Rogers et al. |
|---|---|---|---|---|---|---|---|---|---|---|---|---|
| Authors | Year | 2021 | 2020 | 2020 | 2018 | 2018 | 2010 | 2008 | 2008 | 2007 | 2007 | 1982 |
| Subjects | Male/Female number (%) | 22/18 (55/45) | 21/19 (52.5/47.5) | 14/9 (60.9/39.1) | 18/16 (52.9/47.1) | 11/6 (64.7/35.2) | 30/91 (24.8/75.2) | 37/45 (45.1/54.9) | Unknown | 8/12 (40/60) | 14/5 (73.7/26.3) | 10/8 (55.6/44.4) |
| | Number | 40 | 40 | 23 | 34 | 17 | 121 | 82 | 161 | 20 | 19 | 18 |
| | Age Range (years) | 5–7 | 3–12 | 5–12 | 5.8–17.8 | 5–50 | 12–28 | 10–30 | 3–11 months | Unknown | 4–6 | 4–11 months |
| | Mean age (years) | 5.8±0.6 | 7.3±3.8 | 7.5±2.0 | 9.5±3 | 17.7±14.3 | 18.8 | 8.2±1.7 | 7.1 months | 11.6 months | 5.08±0.59 | 7.6±1.94 months |
| Diagnosis (number) | Strabismus | 40 | 40 | 14 | 16 | 17 | 22 | 45 | 161 | 20 | 19 | 18 |
| | ET | 26 | 40 | Unknown | 9 | 7 | Unknown | Unknown | 161 | 20 | 13 | 18 |
| | XT | 7 | No | Unknown | 7 | 2 | Unknown | Unknown | No | No | No | No |
| | H | 1 | No | Unknown | No | 4 | Unknown | Unknown | No | No | No | No |
| | Combine (ET+H) | 1 | No | Unknown | No | 2 | Unknown | Unknown | No | No | 6 | No |
| | Combine (XT+H) | 5 | No | Unknown | No | 2 | Unknown | Unknown | No | No | No | No |
| | Combined other ophthalmic diseases | No | No | amblyopia (9) | unilateral amblyopia (18) | No | amblyopia (99) | amblyopia (37) | No | No | No | No |
| Stereoacuity (number/%) | Normal | 0 (0) | Unknown | 11 (47.8) | 29 (85.3) | Unknown | Unknown | 5 (6) | Unknown | Unknown | 0 (0) | Unknown |
| | Undetectable | 40 (100) | Unknown | 12 (52.2) | 5 (14.7) | Unknown | Unknown | 77 (94) | Unknown | Unknown | 19 (100) | Unknown |
| Motor Function | Instrument(s) | BOT-2 | MABC-2 | DCDQ | Balance subtest in BOT-2 | 8-camera SMART D motion capture system | Purdue pegboard, bead threading task, and water pouring task | BOTMP | IDSS | BSID | Movement ABC | BSID |
| Results: Normal/Abnormal (%) | BOT-2 CF | 31/9 (77.5/22.5) | | | | | | | | | | |
| | BOT-2 SF | 37/3 (92.5/7.5) | | | | | | | | | | |
| | BOT-2: Balance subtest | 29/11 (72.5/27.5) | | | 3/13 (18.8/81.2) in SG | | | | | | | |

*(Continued)*

Table 1. (Continued)

| Authors | Present study | Hemptinne et al. | Vagge et al. | Zipori et al. | Di Sipio et al. | O'Connor et al. | Webber et al. | Drover et al. | Tukkers-van et al. | Caputo et al. | Rogers et al. |
|---|---|---|---|---|---|---|---|---|---|---|---|
| Year | 2021 | 2020 | 2020 | 2018 | 2018 | 2010 | 2008 | 2008 | 2007 | 2007 | 1982 |
| Movement ABC | | | | | | | | | | Before surgery: Normal: 9 (47.4%), Borderline: 6 (31.6%), Abnormal: 4 (21.1%); After surgery: Normal: 12 (63.2%), Borderline: 4 (21.1%), Abnormal: 3 (15.8%) | |
| MABC-2 | | Normal: 19 (47.5%), Borderline: 13 (32.5%), Abnormal: 8 (20%) | | | | | | | | | |
| BOTMP: VMC; ULSD | | | | | | | 1/44(2.2/ 97.8); 10/ 35(22.2/ 77.8) | | | | |
| DCDQ | | | 17/6 (73.9/ 26.1) | | | | | | | | |
| Purdue pegboard, bead threading task, and water pouring task | | | | | | Fine motor skills were significantly better in children with sensory and motor fusion. | | | | | |
| IDSS | | | | | | | | After surgery: the sensorimotor/ gross milestones were not delayed. | | | |

(Continued)

**Table 1.** (Continued)

| Studies | Authors | Present study | Hemptinne et al. | Vagge et al. | Zipori et al. | Di Sipio et al. | O'Connor et al. | Webber et al. | Drover et at. | Tukkers-van et al. | Caputo et al. | Rogers et al. |
|---|---|---|---|---|---|---|---|---|---|---|---|---|
| | Year | 2021 | 2020 | 2020 | 2018 | 2018 | 2010 | 2008 | 2008 | 2007 | 2007 | 1982 |
| | BSID | | | | | | | | | After surgery: the delay in motor development persisted for months. | | After surgery: an improvement in fine motor skills in 35% of the children. |
| | 8-camera SMART D motion capture system | | | | | After surgery: improvements in mean velocity, cadence, and step length. | | | | | | |

ET: esotropia; XT: exotropia; H: hypertropia; BOT-2: Bruininks-Oseretsky Test of Motor Proficiency-Second Edition; CF: Complete-form; SF: Short-form; MABC-2: Movement Assessment Battery for Children, Second Edition; BOTMP: Bruininks-Oseretsky Test of Motor Proficiency; VMC: Visual motor control; ULSD: upper limb speed and dexterity; DCDQ: Developmental Coordination Disorder Questionnaire; IDSS: Infant Developmental Skills Survey questionnaire; BSID: Bayley Scales of Infant Development; Movement ABC: Movement Assessment Battery for Children; SG: strabismic group.

of preschool children with strabismus is worthy of further discussion. The purpose of this study was to comprehensively investigate motor competency in preschool children with strabismus. Consequently, the consistency of scores obtained using the CF and SF forms of the BOT-2 in preschool children with strabismus was also explored. Finally, possible confounding factors, such as ADHD tendencies and stereopsis were also examined.

## Methods

The Institutional Review Board (IRB) of the Chang Gung Medical Foundation, Lin-Kao, Taiwan reviewed and approved the protocol for this original cohort study before recruitment (IRB No. 201600626B0).

### Participants

All eligible children with strabismus between 5 and 7 years of age who had received services at the department of pediatric ophthalmology in Chang Gung Medical Foundation, Linkou Chang Gung Memorial Hospital were recruited from April 2018 to June 2020. The inclusion criteria were as follows: child aged 5–7 years (Taiwan's Ministry of Education defines children aged 5–7 years as preschool children); correct completion of ophthalmological testing; understanding of test instructions; and informed, written consent. The exclusion criteria were other ophthalmic diagnosis, neurological deficit, genetic disease, intellectual disability, or inability to follow verbal commands (Fig 1). After, preschool children with strabismus and their parents were informed about the purpose and procedure of the study face-to-face. Verbal and written assent and written informed consent were obtained from the children and the guardians, respectively. A total of 40 preschool children (22 boys (55%), 18 girls (45%); mean age = 5.92 ± 0.63 years) with strabismus completed the motor competency assessment.

### Measures

The BOT-2 is a commonly used diagnostic to evaluate the psychometric properties of motor function in children. The BOT-2 CF assesses proficiency in four motor-area composites as follows: the "Fine Manual Control" composite which is divided into a "Fine Motor Precision" subtest (7 items, score range = 0−41 points) and "Fine Motor Integration" subtest (8 items, score range = 0−40 points); the "Manual Coordination" composite which is divided into a "Manual Dexterity" subtest (5 items, score range = 0−45 points) and "Upper-Limb Coordination" subtest (7 items, score range = 0−39 points); the "Body Coordination" composite which is divided into a "Bilateral Coordination" subtest (7 items, score range = 0−24 points) and "Balance" subtest (9 items, score range = 0−37 points), and the "Strength and Agility" composite which is divided into a "Running Speed and Agility" subtest (5 items, score range = 0−52 points) and "Strength" subtest (5 items, score range = 0−42 points). Each raw score on a subtest was converted to a point score to allow performance to be measured on a graded scale. The individual item points were summed to derive a subtest point score. The subtest point scores for eight subtests were summed to achieve a "Total Motor Composite". The "Total Motor Composite" comprising the above motor-area composites provides a measure of the child's overall motor proficiency. The subtest point scores were used to derive a scale score. Scale scores describe comparability to a normative sample of examinees of the same age and have a mean of 15 with a standard deviation of 5. Additionally, scale and standard scores for subtests and composites can be represented in the form of descriptive categories ranging from well below average to well above average [20]. In contrast to the 53 items comprising the total motor composite, the derived BOT-2 SF only includes 14 items. It is used as a screening tool to identify children who need more comprehensive assessment. However, the psychometric

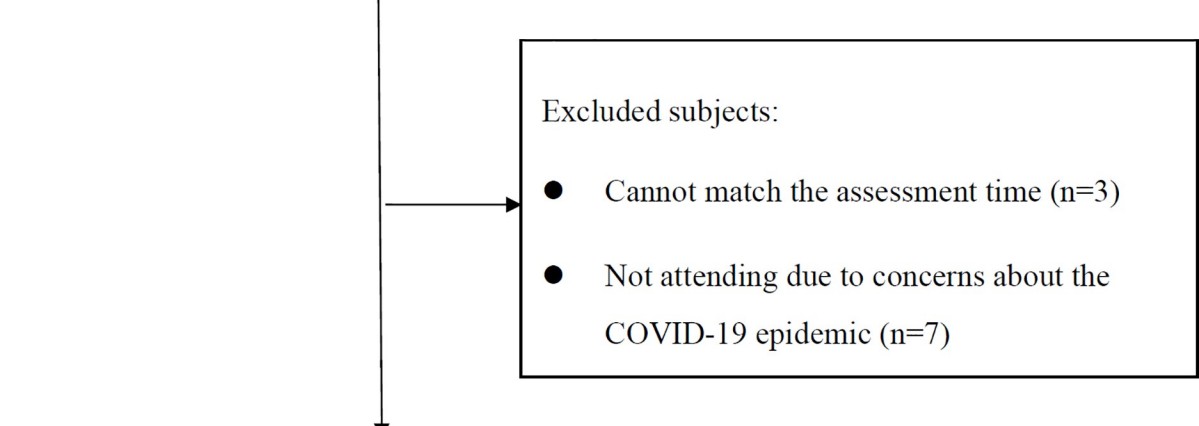

Preschool children with strabismus (n=50):

1. Strabismus diagnosis

2. Five to seven years old (preschool age)

3. Excluded if:

   ■ Combined CNS lesion (e.g., cerebral palsy)

   ■ Unable to follow test instructions

   ■ Diagnosed with otitis media within a month

   ■ Suffered pain caused by the musculoskeletal system within a month

   ■ Combined with other ophthalmological diagnosis (e.g., amblyopia)

Excluded subjects:

● Cannot match the assessment time (n=3)

● Not attending due to concerns about the COVID-19 epidemic (n=7)

● Basic data collection

● Testing for the Bruininks-Oseretsky Test of Motor Proficiency, Second Edition (BOT-2) Complete-form

● Data analysis (n=40)

**Fig 1. Flowchart for inclusion of subjects.**

properties of the BOT-2 SF are purportedly comparable to the BOT-2 CF [20, 21]. Prior to the initiation of this study, the inter-rater reliability of the BOT-2 CF was established by two physical therapists. The first examiner was a licensed pediatric physical therapist with 17 years of clinical experience. Examiner 2 was a licensed physical therapist with 2 years of clinical experience and was enrolled in a master's degree program at the time. Ten videos from the BOT-2 CF were assessed individually. The inter-rater reliability results were as follows: the intraclass correlation coefficient ($ICC_{(2,1)}$) of the "Total point Score", "Scale Score", "Standard Score",

and "Percentile Rank" were good to excellent (0,827–1.000, 0.683–0.982, 0.778–0.956, and 0.836–0.963, respectively.). The kappa of the "Descriptive category" was also good to excellent (0.674–1.000) [25]. The Disruptive Behavior Rating Scale (DBRS) questionnaire is a brief (26-item) rating scale used to assess inattention, hyperactivity–impulsivity, and oppositional defiant behavior in preschool-aged children [26]. The DBRS assesses 18 symptoms of ADHD that indicate inattention and hyperactivity–impulsivity, but also contains 8 additional items to assess oppositional defiant disorder (ODD) symptoms. The cut-off score of inattention is 15 points for boys and 12 points for girls. The cut-off score of hyperactivity–impulsivity is 17 points for boys and 13 points for girls. The use of the DBRS as a measure of disruptive behavior disorder symptoms in preschool-aged children showed strong internal consistency and evidence of convergent/divergent and discriminative validity [27]. This study used 18 items of the DBRS to investigate ADHD tendencies of participants.

## Procedure

Before performing the BOT-2 CF, preschool children with strabismus received an ophthalmologic examination by an ophthalmologist, including cycloplegic refraction examination for best-corrected visual acuity (BCVA) (OD: 1.0: n = 36/40, 90%; 0.9: n = 3/40, 7.5%; 0.8: n = 1/40, 2.5%; OS: 1.0: n = 34/40, 85%; 0.9: n = 5/40, 12.5%; 0.8: n = 1/40, 2.5%) and Titmus test for stereopsis examination (400": n = 9/40, 22.5%; 800": n = 1/40, 2.5%; Not detectable: n = 30/40, 75%). All parents were asked to complete the DBRS for investigating ADHD tendency. BOT-2 CF testing was performed on an individual basis in quiet locations at the Pediatric Physical Therapy (PT) Unit of the Chang Gung Medical Foundation, Linkou Chang Gung Memorial Hospital. All participants were tested by the examiner using the standardized procedures for administration specified in the test manual. Testing lasted approximately one hour, with a suitable number of breaks to minimize the effects of fatigue and frustration.

Preschool children with strabismus were given the BOT-2 CF and were initially classified as "Below Average for Motor Competency" if they scored below the published threshold ($\leq 17^{th}$ %ile rank; Standard Score $\leq 40$), or "Average for Motor Competency" if they scored in the average range or higher ($\geq 18^{th}$ %ile rank; Standard Score $\geq 41$) on either test as per the manual [20].

## Data analysis

All analyses were performed using SPSS (version 25.0; IBM, Armonk, NY, USA). Descriptive statistics were calculated for data analysis (i.e., mean, standard deviation, percentage, and range). The BOT-2 SF was used as a diagnostic test and was compared to the "gold standard" BOT-2 CF. To compare the two tests, the sensitivity, specificity, positive predictive values, negative predictive values, and accuracy were determined with 95% CI [28]. Other cut-offs in the BOT-2 SF were also explored for improved diagnostic accuracy by using receiver-operating characteristics (ROC) curves, area under ROC curve (AUC) [29], and Youden index analysis [30]. Pearson's product-moment coefficients were used to assess whether the 14 items included in the BOT-2 SF test were indeed representative of the respective domains in the BOT-2 CF test [31]. To explore the scores among children with strabismus with different characteristics (ADHD tendency and stereopsis), data were analyzed by t-tests, as appropriate, and statistical significance was defined as $p < 0.05$.

## Results

On the BOT-2 CF, 15% in the "Fine Manual Control" composite, 70% in the "Manual Coordination" composite, 32.5% in the "Body Coordination" composite, and 5% in the "Strength and

Agility" composite of preschool children with strabismus were classified as "Below Average for Motor Competency" (Table 2, Fig 2, S1 and S2 Tables). The "Total Motor Composite" score from the BOT-2 CF and BOT-2 SF were dichotomized, according to the test manual, at the 17[th] %ile rank to yield the following categorical results: 9 of the 40 preschool children with strabismus (22.5%) were classified as "Below Average for Motor Competency" via the BOT-2 CF. In contrast, only 3 of the 40 preschool children with strabismus (7.5%) were correctly identified by the BOT-2 SF (Table 2, Fig 2, S2 Table). Thus, 6 preschool children with strabismus (15%) were misclassified as "Average for Motor Competency" yielding a false-negative rate of 67% (95% CI = 42%–106%) and a sensitivity of 33.33% (95% CI = 7.49%–70.07%).

However, the BOT-2 SF correctly identified all of the "Average for Motor Competency" preschool children with strabismus, yielding a specificity of 100% (95% CI = 88.78%–100%). The positive predictive value was also 100%, indicating that all preschool children with strabismus were positively identified by the BOT-2 CF and the BOT-2 SF (n = 36). The negative predictive value was 83.78% (95% CI = 76.5%–89.13%). The overall accuracy of the BOT-2 SF

**Table 2. BOT-2 complete form (CF) results in preschool children with strabismus.**

| n = 40 | Scale Score | Standard Score | %ile Rank | Descriptive Category numbers (%) | | | | |
|---|---|---|---|---|---|---|---|---|
| | | (Mean/SD) | | WAA | AA | A | BA | WBA |
| Fine Motor Precision | 16.15 ±5.31 | – | – | 3 (7.5%) | 8 (20%) | 25 (62.5%) | 3 (7.5%) | 1 (2.5%) |
| Fine Motor Integration | 13.98 ±4.62 | – | – | 0 (0%) | 7 (17.5%) | 26 (65%) | 5 (12.5%) | 2 (5%) |
| Fine Manual Control | 30.13 ±9.24 | 49.85±10.87 | 49.88 ±29.86 | 1 (2.5%) | 9 (22.5%) | 24 (60%) | 5 (12.5%) | 1 (2.5%) |
| Manual Dexterity | 10.13 ±3.92 | – | – | 0 (0%) | 0 (0%) | 19 (47.5%) | 18 (45%) | 3 (7.5%) |
| Upper-Limb Coordination | 10.05 ±3.17 | – | – | 0 (0%) | 0 (0%) | 14 (35%) | 22 (55%) | 4 (10%) |
| Manual Coordination | 20.18 ±6.16 | 37.88±7.18 | 16.74 ±17.27 | 0 (0%) | 0 (0%) | 12 (30%) | 23 (57.5%) | 5 (12.5%) |
| Bilateral Coordination | 11.68 ±4.65 | – | – | 0 (0%) | 2 (5%) | 24 (60%) | 8 (20%) | 6 (15%) |
| Balance | 14.1 ±5.47 | – | – | 1 (2.5%) | 5 (12.5%) | 23 (57.5%) | 8 (20%) | 3 (7.5%) |
| Body Coordination | 25.78 ±8.48 | 44.82±9.6 | 34.58 ±27.04 | 1 (2.5%) | 1 (2.5%) | 25 (62.5%) | 12 (30%) | 1 (2.5%) |
| Running Speed and Agility | 16.8 ±3.92 | – | – | 1 (2.5%) | 10 (25%) | 27 (67.5%) | 2 (5%) | 0 (0%) |
| Strength | 20.5 ±4.71 | – | – | 7 (17.5%) | 15 (37.5%) | 17 (42.5%) | 1 (2.5%) | 0 (0%) |
| Strength and Agility | 37.3 ±8.01 | 58.18±9.58 | 72.35 ±26.31 | 3 (7.5%) | 18 (45%) | 17 (42.5%) | 2 (5%) | 0 (0%) |
| Total Motor Composite: Complete form | – | 46.95±9.21 | 41.08 ±25.68 | 1 (2.5%) | 2 (5%) | 28 (70%) | 8 (20%) | 1 (2.5%) |
| Total Motor Composite: Short form | – | 52.43±9.53 | 57.18 ±28.67 | 1 (2.5%) | 9 (22.5%) | 27 (67.5%) | 3 (7.5%) | 0 (0%) |

WAA: Well-Above Average (Scale Score≥25; Standard Score≥70; %ile Rank≥98); AA: Above Average (24≥Scale Score≥20; 69≥Standard Score≥60; 97≥%ile Rank≥84); A: Average (19≥Scale Score≥11; 59≥Standard Score≥41; 83≥%ile Rank≥18); BA: Below Average (10≥Scale Score≥6; 40≥Standard Score≥31; 17≥%ile Rank≥3); WBA: Well-Below Average (Scale Score≤5; Standard Score≤30; %ile Rank≤2).

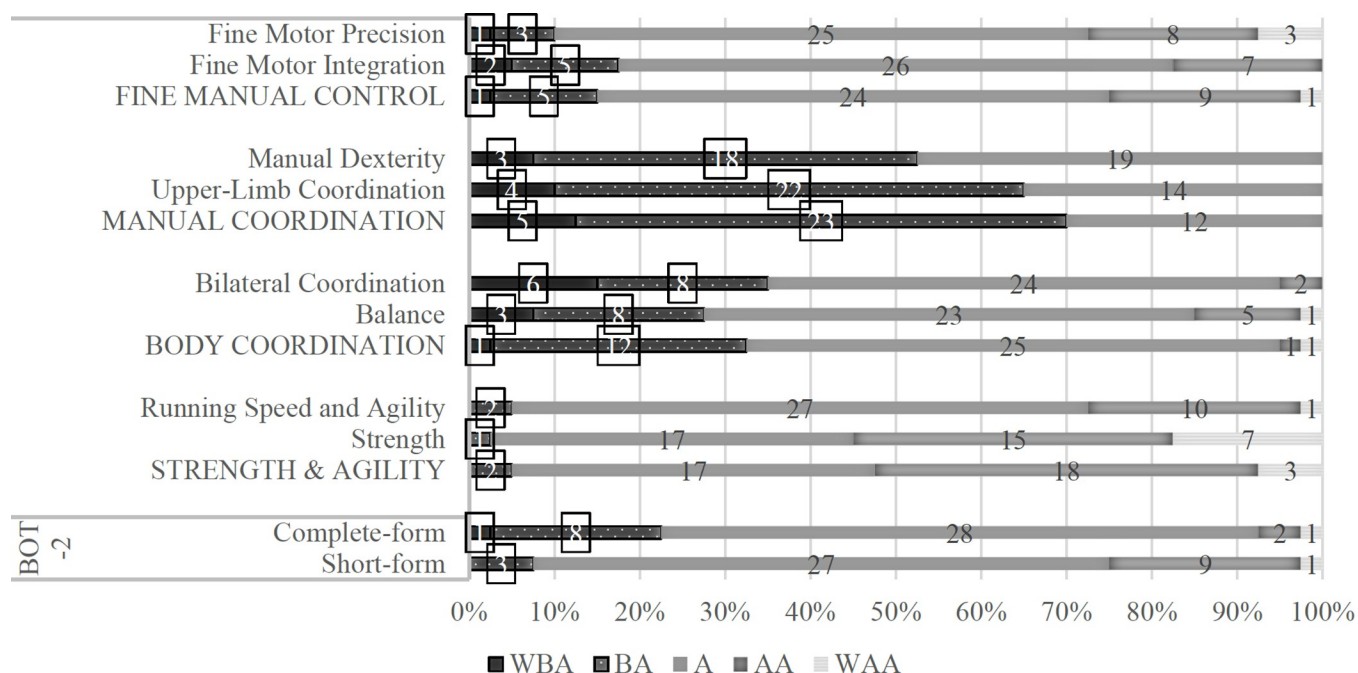

**Fig 2. Distribution of BOT-2 results in preschool children with strabismus.** WBA: Well-Below Average (Scale Score≤5; Standard Score≤30; %ile Rank≤2); BA: Below Average (10≥Scale Score≥6; 40≥Standard Score≥31; 17≥%ile Rank≥3); A: Average (19≥Scale Score≥11; 59≥Standard Score≥41; 83≥%ile Rank≥18); AA: Above Average (24≥Scale Score≥20; 69≥Standard Score≥60; 97≥%ile Rank≥84); WAA: Well-Above Average (Scale Score≥25; Standard Score≥70; %ile Rank≥98).

(i.e., correctly identifying preschool children with strabismus as "Below Average for Motor Competency" or "Average for Motor Competency" as defined by the BOT-2 CF) was 85% (95% CI = 70.16%–94.29%) (Table 3). In terms of ADHD, the DBRS results indicated that boys had an ADHD tendency and girls had a borderline tendency (Table 4).

## Discussion

The main purpose of this study was to comprehensively investigate the motor competency of preschool children with strabismus. One difference from previous studies is that ours recruited preschool children who were only diagnosed with strabismus (without amblyopia); none of the preschool children had normal stereopsis. In the past, no studies had investigated the ability of preschool children with strabismus to manipulate pens and paper. According to our findings, more than half of preschool children with strabismus (n = 34, 85%) were classified as "Average for Motor Competency" via the "Fine Manual Control" composite. The "Fine Manual Control" composite is divided into the "Fine Motor Precision" subtest (e.g., cutting out a circle, connecting dots) and "Fine Motor Integration" subtest (e.g., copying a star, copying) in the BOT-2 CF. All of these test items were performed on the same plane (that is, in two-dimensional space). Therefore, even though the preschool children did not have normal stereopsis, there was still a high percentage of preschool children with strabismus classified as "Average for Motor Competency" in the "Fine Manual Control" composite.

From the results of the present study, the preschool children's strabismus impacted their "Manual Coordination" composite (n = 28/40, 70%) more than the "Fine Manual Control" composite (n = 6/40, 15%) of fine motor competency. These findings are similar to previous studies that focused on the fine motor competency of children with strabismus [12–14, 19]. The test items in the "Manual Dexterity" subtest (e.g., transferring coins, sorting cards,

**Table 3. Evaluation of BOT-2 short form (SF) scores as a diagnostic test using the original (standard score ≤40, or ≤17th %ile rank) and different cut-off scores to identify "Below Average for Motor Competency" (n = 40).**

| Statistic | Short form Standard Score ≤40, or ≤17th %ile Rank (Original cut-off) (20) | Short form Standard Score ≤42, or ≤21st %ile Rank (present study) | Short form Standard Score ≤43.5, ≤25.5th %ile Rank (present study) | Short form Standard Score ≤45.5, ≤33rd %ile Rank (present study suggested cut-off) |
|---|---|---|---|---|
| True positive N | 3 | 4 | 8 | 9 |
| True negative N | 31 | 31 | 30 | 30 |
| False positive N | 0 | 0 | 1 | 0 |
| False negative N | 6 | 5 | 1 | 1 |
| Sensitivity (95%CI) | 33.33% (7.49%–70.07%) | 44.44% (13.7%–78.8%) | 88.89% (71.75%–99.72) | 90% (55.5%–99.75%) |
| Specificity (95% CI) | 100% (88.78%–100%) | 100% (88.78%–100%) | 96.77% (83.3%–99.92%) | 100% (88.43%–100%) |
| Positive predictive value (95% CI) | 100% | 100% | 88.89% (53.43–98.24%) | 100% |
| Negative Predictive value (95% CI) | 83.78 (76.5%–89.13%) | 86.11% (77.56%–91.75%) | 96.77% (82.52%–99.48%) | 96.77% (82.37%–99.48%) |
| Accuracy (95% CI) | 85% (70.16%–94.29%) | 87.5% (73.2%–95.81) | 95% (83.08%–99.39%) | 97.5% (86.84%–99.94%) |

stringing blocks) and in the "Upper-limb Coordination" subtest (e.g., throwing a ball at a target, catching a tossed ball) of the BOT-2 CF are all executed in three-dimensional space, which may present an additional challenge for children with strabismus. Normal stereopsis may provide an important sensory input for the optimal development of fine motor skills in children [32], especially in performing upper limb reaching and grasping movements [33]. Therefore, there was a relatively high proportion of preschool children with strabismus who were judged as "Below Average for Motor Competency" in the "Manual Dexterity" and "Upper-limb Coordination" subtests.

We found that the impact of strabismus on the gross motor competency of preschool children was higher in the "Body Coordination" composite (n = 13/40, 32.5%) compared to the "Strength and Agility" composite (n = 2/40, 5%). Preschool children with strabismus lack the normal binocular vision needed to comprehensively examine the movements demonstrated by the examiner during the "Bilateral Coordination" subtest (e.g., tapping foot and finger,

**Table 4. Results of the Disruptive Behavior Rating Scales (DBRS) questionnaire.**

| DBRS score | Mean ± SD (Range) | Normal | Inattention | Normal | Hyperactive-Impulsive |
|---|---|---|---|---|---|
| | | Numbers (%) | | | |
| Boys (n = 22, 55%) | 18.55±11.37 (4–50) | 10 (45.5%) | 12 (54.5%) | 10 (45.5%) | 12 (54.5%) |
| Girls (n = 18, 45%) | 13±8.43 (2–32) | 9 (50%) | 9 (50%) | 9 (50%) | 9 (50%) |
| Total (n = 40, 100%) | 16.55±10.43 (2–50) | 19 (47.5%) | 21 (52.5%) | 19 (47.5%) | 21 (52.5%) |

DBRS score:
• Boy>15, girl>12, indicate "Inattention".
• Boy>17, girl>13, indicate "Hyperactive-Impulsive".

jumping jacks) of the BOT-2 CF. Patients with strabismus were shown to have a narrower binocular visual field especially those with esotropia [34]. Indeed, the proportion of our preschool cohort with esotropia was relatively high (n = 7/14, 50%) among the 14 children who were judged as "Below Average for Motor Competency" in the "Bilateral Coordination" subtest.

Previous research mainly focused on the parameter of center of gravity (CoG) in different test scenarios (standing on a stable/unstable plane with eyes opened/closed, standing with feet close together/heels on toes etc.) for investigating the balance on the gross motor competency of children with strabismus [35–40]. Only one study focused on functional balance competency in preschool children with strabismus [19]. Caputo and colleagues (2007) found no statistical difference in terms of the functional balance competency (i.e., static balance and dynamic balance) using the Movement Assessment Battery for Children (Movement ABC) between the strabismus group and the control group before and after surgery to correct their strabismus. There were three test items: "One-Leg Balance" (static), "Walk Heels Raised" (dynamic), and "Jumping on Mats" (dynamic) in the balance domain of the Movement ABC. The "Balance" subtest of the BOT-2 CF includes 7 test items (out of 9 total, or 77.8%) focused on static balance, while in the Movement ABC, only 33.3% (1/3) of the test items focused on static balance. Dynamic balance requires continuous postural adjustments to maintain equilibrium thus placing greater demands on the somatosensory and vestibular systems [41]. Therefore, even though strabismus affects the visual system of children, the proportion lagging behind the norm in terms of dynamic balancing tasks was still lower than static balancing tasks for preschool children with strabismus.

The scale scores of the BOT-2 between undetectable and detectable stereopsis in preschool children with strabismus was not significantly different, except on the "Running Speed and Agility" subtest (p = 0.045; ES = 0.071, 95% CI = -0.645, 0.787) (S3 Table). The test items of the "Running Speed and Agility" subtest all challenged the children's dynamic balance ability (e.g., shuttle run, stationary hop, and side hop). The children with detectable stereopsis still had more visual dependence than those with undetectable stereopsis arising from the re-weighting of sensory inputs that has been previously described [42]. As dynamic balance activities rely more on the somatosensory and vestibular system [41], children with undetectable stereopsis may increase the use of the somatosensory and vestibular systems for postural stability compared to children with detectable stereopsis. The performance of "Running Speed and Agility" subtest was therefore better for children with undetectable stereopsis. However, because of the uneven distribution of the two groups of children, it is recommended that further research be done on this topic.

Visual perception influences children's behavior and development, especially attentional abilities, as well as learning and reading processes [43]. Previous research suggests that children with refractive errors and strabismus should be monitored for symptoms of ADHD since this group might have an increased risk for the development of ADHD [24]. Therefore, the DBRS questionnaire was conducted in the present study (Table 4). On the "strength" subtest (p = 0.008; ES = -0.889, 95% CI = -1.540, -0.239) and "Strength and Agility" composite (p = 0.021; ES = -0.762, 95% CI = -1.405, -0.120) of the BOT-2 CF, children without ADHD tendency showed significantly better performance than those with it as judged by the DBRS (S3 Table). The children only needed to maintain 15 seconds for most BOT-2 CF subtests, except for the "Strength" subtest which requires at least 30 seconds or even one minute. This may explain why children with ADHD tendency showed poor activity persistence and subsequent performance on such longer tests. As the calculation of the "Strength and Agility" composite includes the "Strength" subtest, it also led to a better "Strength and Agility" composite for children without ADHD tendency.

The other aim of this study was to evaluate the accuracy of the BOT-2 SF for identification of "Below Average for Motor Competency" preschool children with strabismus relative to the BOT-2 CF. Despite the strong positive correlations between the BOT-2 CF and the BOT-2 SF having been identified previously [20], our findings suggest that discrepancies existed between results obtained via the two tests for preschool children with strabismus. Previous studies identified a tendency for the BOT-2 SF to overestimate a child's motor proficiency compared to the BOT-2 CF [22, 44]. This contributed to a high rate of misclassification of children as not "Below Average for Motor Competency" according to the BOT-2 SF, despite being classified as such via the BOT-2 CF. These findings support previous recommendations that the BOT-2 CF should be used over the BOT-2 SF if a clinician is seeking greater accuracy in their assessment of motor competency of children, particularly if identification of "at-risk" children is important [44].

The test items in the BOT-2 SF may not perfectly recapitulate the broader domains of motor competency assessed in the BOT-2 CF. Earlier validation studies of the BOT-2 have reported that a small number of items retained in the BOT-2 SF are poorly correlated and therefore are not representative of the broader domain that the item is thought to represent in the BOT-2 CF [45, 46]. Pearson's product-moment coefficients were used to assess whether the 14 items included in the BOT-2 SF were indeed representative of the respective domains in the BOT-2 CF. Results revealed significant positive correlations (Pearson's r>0.5) between each item for all corresponding domains [31]. However, there was only a moderate correlation (Pearson's r>0.3) between both "Upper-limb coordination" items included in the BOT-2 SF and the "Upper-limb coordination" captured by the BOT-2 CF (S4 Table) [31]. The false-negative rate was 67% (95% CI = 42%–106%) in the BOT-2 SF in preschool children with strabismus. Therefore, the BOT-2 CF is recommended for evaluating motor competency in preschool children with strabismus, especially for the "Upper-Limb Coordination" subtest.

Jírovec and colleagues (2019) found that the BOT-2 SF had acceptable sensitivity but poor specificity compared to the BOT-2 CF in middle-age school children [47]. Mancini and colleagues (2020) found that the BOT-2 SF had lower sensitivity but flawless specificity compared to the BOT-2 CF in children with ADHD [22]. The results of our study were contrary to Jírovec's study, but similar to Mancini's: the sensitivity was lower (33.33%, 95% CI = 7.49%–70.07%) and the specificity higher (100%, 95% CI = 88.78%–100%) for BOT-2 SF in our study.

We then explored other cut-offs to improve the sensitivity of the BOT-2 SF using ROC curve analysis (Table 3). At the published cut-off (i.e., Standard Score ≤40 or ≤17th %ile rank), the false-negative rate was 66.67%. The AUC statistic was 0.986 (95% CI = 0.954–1.000). The BOT-2 SF was evaluated as a screening tool, prioritizing sensitivity rather than specificity, thereby allowing for more "Below Average for Motor Competency" preschool children with strabismus to be correctly identified, albeit at the cost of higher false positives. Two potential higher cut-offs of the BOT-2 SF for preschool children with strabismus were identified, as summarized in Table 3. At the cut-offs of Standard Score ≤43.5 (i.e., ≤25.5th %ile rank) and Standard Score ≤45.5 (i.e., ≤33rd %ile rank), the false-negative rates were reduced to 11.11% and 10%, respectively, thereby reducing the number of false negatives by 55.56% and 56.67%, respectively. However, the specificity was reduced to 96.77% from 100% at the cut-off of Standard Score ≤43.5.

The results of the present study should be confirmed with a larger sample size for performing subgroup analysis (e.g., on different strabismus types). In addition, the control group, children with strabismus and normal stereopsis, is crucial for future studies to confirm whether the present results are consistent between children in healthy control groups, those with strabismus with normal stereopsis, and those with strabismus without normal stereopsis.

## Conclusions

This study confirms that strabismus has a greater impact on fine rather than gross motor competency in preschool children, especially when performing activities in three-dimensional space. Lack of normal stereopsis and binocular visual field may be contributing factors to impaired motor competency. The BOT-2 is a useful tool to evaluate motor competency in preschool children with strabismus. However, the current findings suggest that motor competency in preschool children with strabismus should be assessed by using the BOT-2 CF rather than the BOT-2 SF wherever possible.

## Supporting information

**S1 Table. Subtests (Fine Motor Precision/Fine Motor Integration/Manual Dexterity/Upper-Limb Coordination) and composite (Fine Manual control/Manual Coordination) results of BOT-2 in preschool children with strabismus.**
(DOCX)

**S2 Table. Subtests (Bilateral Coordination/Balance/Running Speed and Agility/Strength), composite (Body Coordination/ Strength and Agility), and total motor composite of complete-form/short-form results of BOT-2 in preschool children with strabismus.**
(DOCX)

**S3 Table. Results of the BOT-2 between undetectable and detectable stereopsis / with ADHD tendency and without ADHD tendency judged by the disruptive behavior rating scales in preschool children with strabismus.**
(DOCX)

**S4 Table. Correlation between BOT-2 SF items and corresponding domain scores from the BOT-2 CF.**
(DOCX)

## Acknowledgments

The authors would like to thank the parents and their children for their participation.

## Author Contributions

**Conceptualization:** Kuo-Kuang Yeh, Wen-Yu Liu, Meng-Ling Yang, Chun-Hsiu Liu, Hen-Yu Lien, Chia-Ying Chung.

**Data curation:** Kuo-Kuang Yeh, Wen-Yu Liu, Meng-Ling Yang, Chun-Hsiu Liu, Hen-Yu Lien, Chia-Ying Chung.

**Formal analysis:** Kuo-Kuang Yeh, Wen-Yu Liu.

**Funding acquisition:** Wen-Yu Liu.

**Investigation:** Kuo-Kuang Yeh.

**Methodology:** Kuo-Kuang Yeh, Wen-Yu Liu, Meng-Ling Yang, Chun-Hsiu Liu, Hen-Yu Lien, Chia-Ying Chung.

**Project administration:** Kuo-Kuang Yeh, Wen-Yu Liu, Meng-Ling Yang, Chun-Hsiu Liu, Hen-Yu Lien, Chia-Ying Chung.

**Resources:** Meng-Ling Yang, Chun-Hsiu Liu, Chia-Ying Chung.

**Software:** Kuo-Kuang Yeh.

**Supervision:** Wen-Yu Liu, Meng-Ling Yang, Chun-Hsiu Liu, Hen-Yu Lien, Chia-Ying Chung.

**Validation:** Kuo-Kuang Yeh, Wen-Yu Liu, Meng-Ling Yang, Chun-Hsiu Liu, Hen-Yu Lien, Chia-Ying Chung.

**Visualization:** Kuo-Kuang Yeh.

**Writing – original draft:** Kuo-Kuang Yeh.

**Writing – review & editing:** Kuo-Kuang Yeh, Wen-Yu Liu, Meng-Ling Yang, Chun-Hsiu Liu, Hen-Yu Lien, Chia-Ying Chung.

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
