## [Decision Letter · Decision Letter 0]

15 Sep 2021

PONE-D-21-21187Sufficiency of the BOT-2 short form to screen motor competency in preschool children with strabismus.PLOS ONE

Dear Dr. Liu,

Thank you for submitting your manuscript to PLOS ONE. After careful consideration, we feel that it has merit but does not fully meet PLOS ONE’s publication criteria as it currently stands. Therefore, we invite you to submit a revised version of the manuscript that addresses the points raised during the review process. Two expert reviewers have evaluated your manuscript. Both reviewers highlight the merits of your work, and provide simple and constructive suggestions that will help clarify your study. I believe all reviewer comments should be straightforward to address, so I look forward to receiving your revised manuscript. 

We look forward to receiving your revised manuscript.

Kind regards,

Guido Maiello

Academic Editor

PLOS ONE

Journal Requirements:

Reviewers' comments:

Reviewer's Responses to Questions

**Comments to the Author**

1. Is the manuscript technically sound, and do the data support the conclusions?

Reviewer #1: Yes

Reviewer #2: Partly

2. Has the statistical analysis been performed appropriately and rigorously? 

Reviewer #1: Yes

Reviewer #2: Yes

3. Have the authors made all data underlying the findings in their manuscript fully available?

Reviewer #1: No

Reviewer #2: Yes

4. Is the manuscript presented in an intelligible fashion and written in standard English?

Reviewer #1: Yes

Reviewer #2: Yes

5. Review Comments to the Author

Reviewer #1: The article is interesting and well written. Can the authors include a section describing correlation between abnormal scores on BOT-CF and Bot-SF to ADHD scores- in other words break down the scores of motor competency as a function of presence/absence of ADHD and tendency of ADHD.

Reviewer #2: While this paper shows the importance for preschool children with Strabismus and using the BOT-2 CF and SF, I believe this paper would benefit from another thorough read through. Most of the paper should be re-organized with missing information in the various sections. Please check out my specific comments in the document attached.

6. PLOS authors have the option to publish the peer review history of their article (what does this mean?). If published, this will include your full peer review and any attached files.

Reviewer #1: No

Reviewer #2: No

---

## [Author Response · Author response to Decision Letter 0]

27 Oct 2021

Dear Dr. Editor,

We appreciate the comments we have received on our manuscript, "Sufficiency of the BOT-2 short form to screen motor competency in preschool children with strabismus. (Ref. No.: PONE-D-21-21187)." We value the useful and practical suggestions and have revised the manuscript and title according to the reviewers’ suggestions. Additionally, we apologize for our language mistakes. This revised manuscript has undergone English proofreading and editing by two English editing experts. The revised texts are highlighted in red and underlined. In addition, the funders took the role in the study as follows: "the funders had no role in study design, data collection and analysis, decision to publish, but provided partial financial support for the English editing after the completion of first draft of this manuscript." The current funding sources is Chang Gung Memorial Hospital, Linkou (award number: BMRP652, and the recipient: Wen-Yu Liu). Finally, the authors have declared that no competing interests exist. We are sorry for the wrong description of funding and competing interest.

We appreciate your timely review of our work and look forward to hearing from you.

Yours sincerely.

---

## [Decision Letter · Decision Letter 1]

6 Dec 2021

Sufficiency of the BOT-2 short form to screen motor competency in preschool children with strabismus.

PONE-D-21-21187R1

Dear Dr. Liu,

We’re pleased to inform you that your manuscript has been judged scientifically suitable for publication and will be formally accepted for publication once it meets all outstanding technical requirements.

Kind regards,

Guido Maiello

Academic Editor

PLOS ONE

Additional Editor Comments (optional):

Reviewers' comments:

Reviewer's Responses to Questions

**Comments to the Author**

1. If the authors have adequately addressed your comments raised in a previous round of review and you feel that this manuscript is now acceptable for publication, you may indicate that here to bypass the “Comments to the Author” section, enter your conflict of interest statement in the “Confidential to Editor” section, and submit your "Accept" recommendation.

Reviewer #1: All comments have been addressed

Reviewer #2: All comments have been addressed

2. Is the manuscript technically sound, and do the data support the conclusions?

Reviewer #1: Yes

Reviewer #2: Yes

3. Has the statistical analysis been performed appropriately and rigorously? 

Reviewer #1: Yes

Reviewer #2: Yes

4. Have the authors made all data underlying the findings in their manuscript fully available?

Reviewer #1: Yes

Reviewer #2: Yes

5. Is the manuscript presented in an intelligible fashion and written in standard English?

Reviewer #1: Yes

Reviewer #2: Yes

6. Review Comments to the Author

Reviewer #1: The authors have diligently addressed all the concerns raised by the reviewers and the revised manuscript is much improved.

Reviewer #2: (No Response)

7. PLOS authors have the option to publish the peer review history of their article (what does this mean?). If published, this will include your full peer review and any attached files.

Reviewer #1: No

Reviewer #2: No

---

## [Editor Report · Acceptance letter]

9 Dec 2021

PONE-D-21-21187R1 

Sufficiency of the BOT-2 short form to screen motor competency in preschool children with strabismus. 

Dear Dr. Liu:

I'm pleased to inform you that your manuscript has been deemed suitable for publication in PLOS ONE. Congratulations! Your manuscript is now with our production department. 

Kind regards, 

on behalf of

Dr. Guido Maiello 

Academic Editor

PLOS ONE